# Taxonomic and Functional Characteristics of the Gill and Gastrointestinal Microbiota and Its Correlation with Intestinal Metabolites in NEW GIFT Strain of Farmed Adult Nile Tilapia (*Oreochromis niloticus*)

**DOI:** 10.3390/microorganisms9030617

**Published:** 2021-03-17

**Authors:** Zhenbing Wu, Qianqian Zhang, Yaoyao Lin, Jingwen Hao, Shuyi Wang, Jingyong Zhang, Aihua Li

**Affiliations:** 1State Key Laboratory of Freshwater Ecology and Biotechnology, Institute of Hydrobiology, Chinese Academy of Sciences, Wuhan 430072, China; wuzhenbing@ihb.ac.cn (Z.W.); zqq@ihb.ac.cn (Q.Z.); linyaoyao@ihb.ac.cn (Y.L.); haojingwen@ihb.ac.cn (J.H.); wangsy@ihb.ac.cn (S.W.); zhangjy@ihb.ac.cn (J.Z.); 2College of Life Sciences, University of Chinese Academy of Sciences, Beijing 100049, China; 3Key Laboratory of Aquaculture Disease Control, Ministry of Agriculture, Wuhan 430072, China

**Keywords:** Nile tilapia, gill, gastrointestinal tract, microbiota, microbial function, high-throughput sequencing, metabolomics

## Abstract

The gill and gastrointestinal tract are primary entry routes for pathogens. The symbiotic microbiota are essential to the health, nutrition and disease of fish. Though the intestinal microbiota of Nile tilapia (*Oreochromis niloticus*) has been extensively studied, information on the mucosa-associated microbiota of this species, especially the gill and gastrointestinal mucosa-associated microbiota, is lacking. This study aimed to characterize the gill and gastrointestinal mucosa- and digesta-associated microbiota, as well as the intestinal metabolite profiles in the New Genetically Improved Farmed Tilapia (NEW GIFT) strain of farmed adult Nile tilapia by high-throughput sequencing and gas chromatography/mass spectrometry metabolomics. The diversity, structure, composition, and predicted function of gastrointestinal microbiota were significantly different across gastrointestinal regions and sample types (Welch *t*-test; *p* < 0.05). By comparing the mucosa- and digesta-associated microbiota, linear discriminant analysis (LDA) effect size (LEfSe) analysis revealed that *Pelomonas*, *Ralstonia*
*pickettii*, Comamonadaceae, and *Staphylococcus* were significantly enriched in the mucosa-associated microbiota, whereas many bacterial taxa were significantly enriched in the digesta-associated microbiota, including Chitinophagaceae, *Cetobacterium*, *Candidatus*
*Competibacter*, *Methyloparacoccus*, and chloroplast (LDA score > 3.5). Furthermore, *Undibacterium*, *Escherichia*-*Shigella*, *Paeniclostridium*, and *Cetobacterium* were dominant in the intestinal contents and mucosae, whereas *Sphingomonas*
*aquatilis* and *Roseomonas*
*gilardii* were commonly found in the gill and stomach mucosae. The Phylogenetic Investigation of Communities by Reconstruction of Unobserved States (PICRUSt2) analysis revealed that the predictive function of digesta-associated microbiota significantly differed from that of mucosa-associated microbiota (R = 0.8152, *p* = 0.0001). In addition, our results showed a significant interdependence between specific intestinal microbes and metabolites. Notably, the relative abundance values of several potentially beneficial microbes, including *Undibacterium*, *Crenothrix*, and *Cetobacterium*, were positively correlated with most intestinal metabolites, whereas the relative abundance values of some potential opportunistic pathogens, including *Acinetobacter*, *Mycobacterium*, *Escherichia*-*Shigella*, *Paeniclostridium*, *Aeromonas*, and *Clostridium*
*sensu*
*stricto* 1, were negatively correlated with most intestinal metabolites. This study revealed the characteristics of gill and gastrointestinal mucosa-associated and digesta-associated microbiota of farmed Nile tilapia and identified a close correlation between intestinal microbes and metabolites. The results serve as a basis for the effective application of targeted probiotics or prebiotics in the diet to regulate the nutrition and health of farmed tilapia.

## 1. Introduction

The microbiota of fish, like other vertebrates, play critical roles in host health and metabolism [1]. Fish microbiota include resident communities (autochthonous or associated with mucosa) and transient communities (allochthonous or associated with digesta) [2]. Most infections start at or affect the fish mucosal epithelia. The three major infection routes in fish are the gills, the skin, and the gastrointestinal tract [3], which are the first points of contact for pathogens. The mucosal surfaces of these organs represent an important barrier that supports and regulates a wide variety of microbial communities [2]. These diverse and balanced mucosa-associated microbial communities are essential for fish health [4], but the mechanism underlying their interactions with the host remains unclear. 

Characterizing the microbiota in healthy fish is an essential first step to elucidating the impacts of microbial manipulation in aquaculture systems. The evaluation of the bacterial community attached to the gills, which is the respiratory organ, has a long history [5]. However, except for several recent studies [6,7,8,9,10], this topic has received little attention. The gastrointestinal tract is a complex ecosystem that harbors diverse microbial communities that can increase digestion efficiency and the use of nutrients, boost the immune system, and prevent the attachment and proliferation of opportunistic pathogens [11]. The establishment of balanced gastrointestinal microbiota is important in fish health and digestive function [11]. However, information regarding the functional roles that these microbes play in host metabolism is limited. 

The New Genetically Improved Farmed Tilapia (NEW GIFT) strain of Nile tilapia is the most commonly farmed tilapia in China and Southeast Asia [12]. The great impact and success of NEW GIFT is evident in developing countries, in which it has helped improve food and income security [12]. However, intensive aquaculture of tilapia has caused various disease problems, especially bacterial diseases [13], which has ultimately led to considerable mortality and consequent economic losses. Recently, insights into the farmed fish microbiome have opened up tantalizing new prospects for optimizing health and productivity in aquaculture systems, such as an increase in the application of probiotics [14]. Host-associated probiotics have a greater chance of competing with resident microorganisms, thus more effectively colonizing the host [15]. Therefore, the use of host-associated probiotics for disease prevention has become a consensus in aquaculture [15], and related studies have been conducted in tilapia [16,17,18]. To better screen host-associated probiotics, the gill and gastrointestinal mucosa-specific microbiota of farmed tilapia need to be further studied.

To date, knowledge of the fish normal microbiome has mostly focused on the characteristics of microbial diversity, and little information on the functional capacity of the fish microbiome is available [19]. In the present study, we used 16S rRNA gene high-throughput sequencing and non-targeted gas chromatography/mass spectrometry (GC/MS) metabolomics technology to characterize the taxonomic diversity and functional potential of a farmed tilapia microbiome.

The objectives of this study were as follows: (1) to characterize the structure and diversity of the gill and gastrointestinal digesta- and mucosa-associated microbiota in the NEW GIFT strain of farmed adult Nile tilapia, (2) to compare the predicted function of symbiotic microbiota at different sites in NEW GIFT Nile tilapia, and (3) to explore the correlation between intestinal microbiota and metabolites in NEW GIFT Nile tilapia under commercial aquaculture conditions. Results provide basic support for the nutritional health and disease prevention of farmed tilapia in aquaculture.

## 2. Materials and Methods

### 2.1. Sample Collection and Preparation

Nile tilapia (NEW GIFT strain of *Oreochromis niloticus*) was sourced from the same batch of artificially incubated seeds and raised in flow-through aquaculture systems at Yingshan County, Huanggang City, Hubei Province, China. The experimental water was from a warm spring in the local mountain, and the water quality was fresh and unpolluted. After oxygenation and purification treatment in the aeration tank, all water quality parameters were within the normal range for tilapia. The water temperature was kept at about 28.5 °C, which was within the optimal temperature range for tilapia farming [20,21]. The fish were fed with the commercial diet at a daily rate of 3% body weight during the farming stage. The feed was provided by Huai’an TianShen Feed Co., Ltd., China (crude protein ≥ 30.0%, crude fat ≥ 4.5%, crude fiber ≥ 9.0%, crude ash ≥ 12.0%, total phosphorus ≥ 0.8%, sodium chloride 0.4–4.0%, lysine ≥ 1.7%, and moisture ≤ 12.5%). No disease symptoms were observed in the fish for 3 months before and after sampling.

Twelve adult tilapias with nearly same size (mean weight, 0.24 ± 0.02 kg) were randomly sampled from four ponds (three fish from each pond) on 20 October 2016. The four ponds had very similar characteristics, including fish density, weight, and the exact same date of introduction of juveniles. The sampled fish were anesthetized using an MS-222 (Sigma, Darmstadt, Germany) solution and then washed with 70% ethanol to reduce contamination before dissection. Under sterile conditions, the gill filaments in the middle of branchial arch were collected and washed thrice in sterile 0.9% saline. Then, the stomach and distal intestines (hindgut) were removed from the abdominal cavity, and their contents were collected. Thereafter, the stomachs and distal intestinal segments were washed using sterile 0.9% saline to remove any leftover contents. The stomach mucosae were scraped with sterile scissors into sterile centrifuge tubes. The intestinal segments and gill filaments were cut into pieces with sterile scissors. To obtain a sufficient bacterial DNA concentration and minimize individual differences [22,23], the gill filaments (G), stomach contents (S), stomach mucosae (W), intestinal contents (C), and intestinal mucosae (M) of three fish from the same pond were pooled and homogenized as one sample (four replications per sample type). The gill filaments, stomach mucosae, and intestinal mucosae were collectively regarded as the mucosa samples. The stomach and intestinal contents were collectively regarded as the digesta samples. All samples were snap frozen in liquid nitrogen and kept frozen at −80 °C.

### 2.2. DNA Extraction, PCR Amplification, and Miseq Sequencing

Total genomic DNA was extracted from the pooled samples using the QIAamp^®^ DNA Stool Mini Kit (Qiagen, Hilden, Germany) according to the manufacturer’s instructions. The purity and concentration of the extracted DNA were checked using a NanoDrop 2000 spectrophotometer (Thermo Fisher Scientific, Waltham, MA, USA). The extracted DNA was diluted to 10 ng μL^−1^ and stored at −80 °C for downstream research. The primers 515F (5’-GTGCCAGCMGCCGCGGTAA-3’) and 909R (5’-CCCCGYCAATTCMTTTRAGT-3’) with a 12 nt unique barcode at 5′-end of 515F were used to amplify the V4–V5 hypervariable region of the 16S rRNA gene [24]. The PCR amplification was conducted in duplicate with a PCR mixture (25 μL) that contained 1 × PCR buffer, 1.5 mM MgCl_2_, each deoxynucleoside triphosphate at 0.4 μM, each primer at 1.0 μM, and 0.5 U of Ex Taq (TaKaRa, Dalian, China) and 10 ng DNA template. The PCR amplification program included an initial denaturation at 94 °C for 3 min, followed by 30 cycles of 94 °C for 40 s, 56 °C for 60 s, 72 °C for 60 s, and a final extension at 72 °C for 10 min.

After PCR amplification, the PCR products were purified using a DNA Gel Extraction Kit (Sangon Biotech, Shanghai, China) and pooled with equimolar quantity. The TruSeq^®^ DNA PCR-Free Sample Preparation Kit was used to construct the library according to manufacturer’s instructions. After quantification by Qubit and qPCR, amplicon libraries were sequenced using the Illumina Miseq platform at the Environmental Genome Platform of Chengdu Institute of Biology, Chinese Academy of Sciences.

### 2.3. Extraction and Detection of Metabolites

Metabolite extraction and detection from the intestinal content samples were performed according to previous studies [25] with some modifications. Fresh intestinal content samples at 100 mg were ground using liquid nitrogen and transferred into 5 mL centrifuge tubes. Then, 500 μL of ddH_2_O (4 °C) were added and vortexed for 60 s, followed by 1000 μL of methanol (pre-cooled at −20 °C), 60 μL of 2-chloro-L-phenylalanine (0.2 mg/mL stock in methanol), and 60 μL of heptadecanoic acid (0.2 mg/mL stock in methanol) as the internal quantitative standard and vortex for 30 s. The tubes were placed into an ultrasound machine at room temperature for 10 min and then stewed for 30 min on ice. The tubes were centrifuged for 10 min at 12,000 rpm (4 °C), and 1.2 mL of the supernatant was transferred into a new centrifuge tube. Samples were blow-dried by vacuum concentration. A methoxyamine pyridine solution (60 μL of 15 mg/mL) was then added. The solution was vortexed for 30 s and reacted for 120 min at 37 °C. Then, 60 μL of a BSTFA reagent (containing 1% TMCS) were added into the mixture and reacted for 90 min at 37 °C. The mixture was centrifuged for 10 min at 12,000 rpm (4 °C), and the supernatant was transferred to an inspection bottle. For the quality control (QC) samples, 20 µL from each prepared sample extract was mixed. These QC samples were used to monitor deviations of the analytical results from these pool mixtures and to compare them to the errors caused by the analytical instrument itself. Finally, the rest of the samples were used for GC-MS test detection.

After the above reactions, samples were determined for metabolites using an Agilent 7890A GC system coupled to an Agilent 5975C inert XL EI/CI mass spectrometric detector (MSD) system (Agilent Technologies, Santa Clara, CA, USA). Gas chromatography was performed on a HP-5MS capillary column (5% phenyl/95% methylpolysiloxane 30 m × 250 μm i.d., 0.25 μm film thickness, Agilent J & W Scientific, Folsom, CA, USA) to separate the derivatives at a constant flow of 1 mL/min helium. Then, 1 µL of the sample was injected in split mode in a 20:1 split ratio by the auto-sampler. The injection temperature was 280 °C. The interface set to 150 °C, and the ion source was adjusted to 230 °C. The programs of temperature-rise were followed by an initial temperature of 60 °C for 2 min at 10 °C/min rate up to 300 °C and maintained at 300 °C for 5 min. Mass spectrometry was determined by full-scan method within the range of 35–750 (m/z).

### 2.4. Metabolite Profiling Analysis

Raw gas chromatography/mass spectrometry (GC/MS) data were converted into the netCDF (network Common Data Form) format (namely XCMS input file format) via an Agilent MSD ChemStation workstation [26]. The XCMS (www.bioconductor.org; 8 June 2017) package in the R software (v3.1.3) was used to conduct peak identification, filtration, and alignment. The data matrixes, including the mass to charge ratio (m/z), retention time, and intensity, were obtained. The annotation of metabolites using the Automatic Mass Spectral Deconvolution and Identification System (AMIDS) was searched against commercially available databases, such as National Institute of Standards and Technology (NIST) and Wiley Registry Metabolomics Database. The alkane retention indices provided by the Golm Metabolome Database (GMD) (http://gmd.mpimp-golm.mpg.de/; 8 June 2017) were used for the further qualitative characterization of substances. Most substances were further confirmed by the standard. The data were derived to Microsoft Excel (Microsoft, Redmond, WA, USA). Finally, the data were normalized to the internal standard for further statistical analyses.

### 2.5. Sequencing Data Processing

The paired-end reads from the raw DNA fragments were merged by FLASH software and quality-filtered by Trimmomatic with the following data decontamination methods and parameters. (1) The 300 bp reads were truncated at any site receiving an average quality score <20 over a 50 bp sliding window, and then the truncated reads that were shorter than 50 bp were discarded. Reads containing N-bases were also removed. (2) The pair-end reads were merged into a sequence according to their overlap relationship, and the minimum overlap length was 10 bp. (3) The maximum mismatch ratio allowed in the overlap area of the merged sequence was 0.2. (4) Samples were distinguished and the direction of the sequence was corrected based on the barcode and primer sequences at both ends of the sequence. The number of mismatches allowed in the barcode was 0, and the maximum number of mismatched primers was 2. Operational taxonomic units (OTUs) were clustered with a 97% similarity cut-off using UPARSE (version 7.0 http://drive5.com/uparse/; accessed on 27 August 2020), and chimeric sequences were identified and removed using UCHIME. The taxonomy of each OTU representative sequence was analyzed by an RDP classifier (http://rdp.cme.msu.edu/; accessed on 27 August 2020) against the Bacterial Silva 16S rRNA database (SILVA SSU 138).

### 2.6. Bioinformatics and Statistical Analyses

Sequences from each sample were randomly resampled to 4098 reads based on the minimum number of valid sequences in the samples to eliminate the effect of sequencing depth on subsequent analyses. Alpha-diversity was estimated with the richness indices of the observed richness (OTUs) and Chao, the diversity indices of Shannon and Simpson, and Good’s coverage (coverage). Beta-diversity was analyzed with principal coordinates analysis (PCoA) and analysis of similarity (ANOSIM) based on the Bray-Curtis metric. The linear discriminant analysis (LDA) effect size (LEfSe) is an algorithm used for the high-dimensional biomarker discovery and explanation that identify genomic features characterizing the differences under two or more biological conditions [27]. In the present study, an LEfSe analysis was performed to identify microbial biomarkers and functional differences with the alpha parameter of 0.05 and an LDA threshold value of 3.5. Differences between two independent groups were evaluated using the Welch’s *t*-test (Past, version 3.15) [28]. The *p*-value was corrected using the Bonferroni method.

The Phylogenetic Investigation of Communities by Reconstruction of Unobserved States (PICRUSt) is widely used to predict microbial functions and metabolic pathways [29]. PICRUSt2 predictions based on several gene family databases, including the Kyoto Encyclopedia of Genes and Genomes (KEGG) orthologs and Enzyme Commission numbers, were supported by default [30]. In the present study, the functional profiles of the bacterial communities were predicted using the PICRUSt2 from the KEGG pathways. The accuracy of the functional predictions was assessed through the computation of the Nearest Sequenced Taxon Index (NSTI) and decreased with increasing NSTI value [29]. The statistical analysis of the microbial function was performed using the Statistical Analysis of Metagenomics Profiles [31]. Data were expressed as mean ± standard deviation (*n* = 4), and the significance level of the difference was set at 0.05 or 0.01.

## 3. Results

### 3.1. Diversity and Structure of the Bacterial Communities

After the quality filtering process, 262,861 valid reads (ranging from 4098 to 31,832 per sample) were obtained from all 20 samples. Then, after random resampling, the reserved sequences were clustered into a total of 487 OTUs. The Good’s coverage ranged from 98.58% to 99.78% (99.43 ± 0.40%; Appendix A). The rarefaction curves tended to the plateau level, and the Shannon curves were stable (Appendix A). These results indicated that the majority of the microbial diversity present in the samples was detected.

Statistical analysis showed that no significant differences (Welch *t*-test; *p* > 0.05) were found in the richness and diversity of the bacterial communities among the three types of mucosa samples (G, M, and W), as evaluated with the richness estimators of OTUs and Chao 1 and with the diversity indices of Shannon and Simpson (Figure 1A–D; Appendix A). Notably, the richness and diversity indices showed that the bacterial communities of the mucosa samples had significantly lower richness and diversity values than those of the digesta samples (C and S) (Figure 1A–D; Appendix A). Furthermore, the bacterial communities of the stomach contents were significantly richer and more diverse than those of the intestinal contents (Figure 1A–D; Appendix A).

Multivariate statistical analyses were conducted to compare the integral structure of the bacterial communities in different sites. ANOSIM revealed significant differences (*p* = 0.034) in the bacterial community structures between any two gastrointestinal sites (Appendix A). The PCoA plot visualized the ANOSIM results, which showed distinct separations of the bacterial communities among four gastrointestinal sites (Figure 2A). Overall, the two principal coordinates obtained from the PCoA explained 59.56% of the variations among all samples. Interestingly, no significant difference (R = −0.0417, *p* = 0.497) was observed in the bacterial communities between the gill and stomach mucosae (Appendix A and Figure 2A). The hierarchical clustering tree on OTU level disclosed that the bacterial communities in the intestinal content samples clustered into one branch first and then clustered with the stomach content samples, whereas the gills and stomach mucosa samples clustered together into another branch (Figure 2B).

### 3.2. Taxonomic Composition of the Bacterial Communities

The phylogenetic classification of sequences from all samples was assigned to 27 bacterial phyla, 64 classes, 132 orders, 184 families, 258 genera, and 487 OTUs (Appendix A). At the phylum level, Proteobacteria was the most abundant phylum in all sites, accounting for 33.00%–95.35% of the total classified sequences (Figure 3A). Actinobacteria and Firmicutes were the second major phyla in the gill mucosae, intestinal contents, and stomach mucosae. The stomach contents had Bacteroidota and Cyanobacteria, whereas the intestinal mucosae had Firmicutes, Fusobacteria, and Deinococcota. In addition, by comparing the mucosa samples with the digesta samples, we found that Proteobacteria was significantly enriched (*p* < 0.05) in the mucosa samples, whereas Bacteroidota, Cyanobacteria, Verrucomicrobia, and Chloroflexi were significantly enriched (*p* < 0.05) in the digesta samples (Appendix A). Notably, the relative abundance values of Firmicutes present in the intestinal content samples were significantly higher than those in the stomach content samples (Welch *t*-test; *p* < 0.05).

At the genus level, the bacterial communities in the gill and stomach mucosae were dominated by *Sphingomonas* and *Ralstonia*, followed by unclassified Comamonadaceae, *Pelomonas*, *Methylobacterium*-*Methylorubrum*, *Amnibacterium*, and *Roseomonas* (Figure 3B). The genera *Undibacterium*, *Escherichia*-*Shigella*, *Paeniclostridium*, and *Cetobacterium* were predominant in all intestinal samples, including intestinal contents and mucosae (Figure 3B). The stomach contents had the maximum number of bacterial taxa, in which the dominant taxa included no-rank Chitinophagaceae, no-rank chloroplast, unclassified Comamonadaceae, and no-rank Verrucomicrobiae, followed by unclassified Rhodocyclaceae and *Cetobacterium* (Figure 3B).

### 3.3. Differences of the Bacterial Communities at Different Sites

The co-occurrence network analysis on the OTU level (relative abundance ≥ 0.5%) differentiated the microbiota among the five sites. The bacterial communities in the gill and stomach mucosae were separate from those in the intestinal contents, mucosae, and stomach contents. Most dominant OTUs in the bacterial communities between the gill and stomach mucosae were shared, suggesting that these two sites had similar core species, including *Sphingomonas aquatilis*, *Ralstonia pickettii*, and *Roseomonas gilardii*, as well as unclassified species belonging to Comamonadaceae, *Methylobacterium*-*Methylorubrum*, *Pelomonas*, *Amnibacterium*, and *Staphylococcus* (Appendix A and Appendix A). The core species in the stomach contents belonged to Chitinophagaceae, Verrucomicrobiae, chloroplast, Comamonadaceae, *Novosphingobium*, and *Cetobacterium*. Though the intestinal contents and mucosae had distinct core species, some shared bacterial species were found between these two sites, including bacterial species belonging to *Undibacterium*, *Escherichia*-*Shigella*, and *Paeniclostridium* (Appendix A and Appendix A).

We further confirmed the presence of different OTUs in different sites by LEfSe. LEfSe identified 35 discriminative features (LDA score > 3.5) between the stomach contents and mucosae, in which 28 OTUs was significantly enriched in the stomach contents, including the most dominant Chitinophagaceae sp. OTU342, Verrucomicrobiae sp. OTU360, Comamonadaceae sp. OTU235, chloroplast sp. OTU481, *Novosphingobium* OTU331, and *Cetobacterium* OTU126 (Figure 4A). Conversely, only seven OTUs were significantly enriched in the stomach mucosae, as follows: *Sphingomonas aquatilis* OTU498 and OTU488, *Methylobacterium*-*Methylorubrum* OTU491, *R. gilardii* OTU497, *Amnibacterium* OTU161 and OTU484, and *Staphylococcus* OTU499 (Figure 4A). Regarding the intestinal content samples, *Escherichia*-*Shigella* OTU206 and *Undibacterium* OTU173 were significantly enriched in the intestinal mucosae, whereas some OTUs were significantly enriched in the intestinal contents, including several OTUs from Gammaproteobacteria and *Candidatus Competibacter*, *Cetobacterium* OTU126, Rhizobiales Incertae Sedis sp. OTU75, Arenicellaceae sp. OTU10, and *Sarcina* OTU76 (Figure 4B). By comparing two types of digesta samples, we found that seven OTUs belonging to *Undibacterium*, *Paeniclostridium*, SZB30 (an order affiliated to Gammaproteobacteria), *Candidatus Competibacter,* and Rhizobiales Incertae Sedis were significantly enriched in the intestinal content samples, whereas 23 dominant OTUs were significantly enriched in the stomach content samples, including OTUs belonging to Chitinophagaceae, Verrucomicrobiae, Comamonadaceae, *Novosphingobium*, and chloroplast (Figure 5A). By comparing the mucosa samples with the digesta samples, we found that five OTUs belonging to *Pelomonas*, *R. pickettii*, Comamonadaceae, and Staphylococcus were significantly enriched in the mucosa samples, whereas eleven OTUs were significantly enriched in the digesta samples, including OTUs belonging to Chitinophagaceae, Rhizobiales Incertae Sedis, chloroplast, HOC36, *Cetobacterium*, SZB30, *Methyloparacoccus*, and *Candidatus Competibacter* (Figure 5B).

The gill mucosae were characterized by a preponderance of Comamonadaceae sp. OTU486, *R. pickettii* OTU490, *S. aquatilis* OTU488, and unclassified bacteria OTU160 (Appendix A). The stomach contents were characterized by a preponderance of chloroplast sp. OTU481 and *Cyanobium* PCC 6307 OTU374 from Cyanobacteria, as well as *Clostridium sensu stricto* 1 OTU344, whereas the stomach mucosae were characterized by a preponderance of *R. pickettii* OTU502 and *Staphylococcus* OTU499 (Appendix A). The intestinal contents were characterized by a preponderance of HOC36 sp. OTU65, OTU15, and OTU478; SZB30 sp. OTU87 and OTU148; and *Cetobacterium* OTU126 and *Sarcina* OTU147; meanwhile, the intestinal mucosae were characterized by a preponderance of *Escherichia*-*Shigella* OTU206 and *Aeromonas* OTU199 (Appendix A).

### 3.4. Intestinal Metabolite Profile and Its Correlation with Intestinal Microbiota

In this study, 95 different metabolites were detected in the intestinal contents, including amino acids, lipids, carbohydrates, nucleotides, and vitamins, by means of GC/MS analysis. The metabolite profiles are presented in Appendix A. The metabolite profiles of intestinal contents were dominated by phosphoric acid and leucine, followed by isoleucine, lactic acid, glutamic acid, and glycerol. Spearman correlation heatmaps showed significant correlations (*p* < 0.05) between the intestinal bacterial genera and metabolites (Figure 6A,B and Figure 7A,B). *Mycobacterium* and *Desulfomonile* were negatively correlated with changes in ornithine, glycine, alanine, proline, 9,12-(Z,Z)-octadecadienoic, glycerol, methyl-inositol, succinic acid, 3-hydroxypyridine, and adenine. *Alsobacter*, *Acinetobacter*, *Cyanobium* PCC-6307, and *Turicibacter* were negatively correlated with changes in 2-hydroxyglutaric acid, homoserine, 2-oxoisocaproic acid, glucose, uridine, inosine, and pantothenic acid, but they were positively correlated with changes in rhamnose. *Escherichia*-*Shigella* and *Paeniclostridium* were negatively correlated with changes in malonic acid, 2-ketoglutaric acid, hexadecanoic acid, phosphoric acid, 1-monooctadecanoylglycerol, monomethylphosphate, 9-(Z)-hexadecenoic acid, tetradecanoic acid, cholesterol, erythronic acid, maltose, mannose, xylose, xylitol, threonic acid, ribose, ribitol, uracil, nicotinic acid, 1,3-di-tert-butylbenzene, 2,4,6-tri-tert-butylbenzenethiol, and benzoic acid. *Aeromonas* were negatively correlated with changes in pyroglutamic acid, 4-hydroxyproline, 1-monohexadecanoylglycerol, myo-inositol, myo-inositol-1-phosphate, glycerol-3-phosphate, fumaric acid, 9,12,15-(Z,Z,Z)-octadecatrienoic acid, eicosanoic acid, glyceric acid-3-phosphate, glycolic acid, malic acid, glucose-6-phosphate, fructose-6-phosphate, sorbitol-6-phosphate, sucrose, and threitol. *Syntrophus* were negatively correlated with changes in N-acetylglutamic acid, 4-aminobutyric acid, octadecanoic acid, and arachidonic acid but positively correlated with changes in cysteine, aspartic acid, and methionine. *Sarcina* and *Pirellula* were positively correlated with changes in beta-alanine but negatively correlated with changes in 2-amino-butyric acid. *Undibacterium* was positively correlated with changes in ornithine, glycine, alanine, proline, 9,12-(Z,Z)-octadecadienoic acid, glycerol, methyl-inositol, succinic acid, 3-hydroxypyridine, and adenine. *Crenothrix* and *Cetobacterium* were positively correlated with changes in urea, glyceric acid, citric acid, fructose, pyruvic acid, mannitol, glucaric acid, and gluconic acid, but they were negatively correlated with changes in ribonic acid. *Clostridium sensu stricto* 1 was negatively correlated with changes in docosahexaenoic acid. *Romboutsia*, *Methylocaldum*, *Desulfobacca*, and *Methylocystis* was negatively correlated with changes in lactic acid (Figure 6A,B and Figure 7A,B). The findings revealed a significant interdependence between intestinal metabolites and microorganisms.

### 3.5. Functional Prediction of the Bacterial Communities

PICRUSt2 analysis was conducted to predict the microbial functions of the bacterial microbiota. The mucosa samples had significantly lower (Welch *t*-test; *p* < 0.05) NSTI values than the digesta samples (Appendix A). The accuracy of the PICRUSt2 in predicting the microbial function of fish mucosae was significantly higher than that of fish digesta. For comparison, Human Microbiome Project samples had the lowest NSTI values (0.03 ± 0.02), whereas the hypersaline mat microbiome samples had the highest NSTI values (0.23 ± 0.07) [29]. Thus, the PICRUSt2 had a high accuracy in predicting the microbial function of fish mucosa samples. ANOSIM revealed that that no significant difference (*p* > 0.05) was found between gill and stomach mucosae, but significant differences (*p* < 0.05) were found in the predictive function of the bacterial communities between any two sites (Appendix A).

Based on the KEGG pathways, the bacterial communities in all the samples were enriched with functional categories related to metabolism (65.64%), environmental information processing (9.71%), genetic information processing (8.90%), and cellular processes (8.08%) (Figure 8). At KEGG level 2, a large proportion of microbial functions belonged to carbohydrate metabolism, global and overview maps, amino acid metabolism, energy metabolism, metabolism of cofactors and vitamins, membrane transport, and signal transduction (Figure 8). The functional clustering analysis based on average neighbor (unweighted pair-group method with arithmetic means (UPGMA)) showed that the gill and stomach mucosa samples clustered together into one branch and then clustered with the intestinal mucosa samples, whereas the digesta samples, including the stomach and intestinal contents clustered together into another branch except for one sample (Figure 8). Further ANOSIM revealed significant differences (R = 0.8152, *p* = 0.0001) in the microbial functions between the mucosa and digesta samples. The PCA plot disclosed that the microbial functions in the mucosa samples were separated from those in the digesta samples, which were primarily separated by the PC1 axis, accounting for 59.5% of the variation (Appendix A). By comparing the mucosa samples with the digesta samples, some microbial functions were found to be significantly enriched in the digesta samples, including global and overview maps, nucleotide metabolism, metabolism of cofactors and vitamins, metabolism of terpenoids and polyketides, translation, folding, sorting and degradation, transcription, and the immune system (Figure 9). Conversely, the predictive functions related to the metabolism of other amino acids, xenobiotics biodegradation and metabolism, signal transduction, cell motility, the circulatory system, environmental adaptation, and human diseases were significantly enriched in the mucosa samples (Figure 9).

## 4. Discussion

The complex microbiota of fish have received much attention due to their important role in fish health [19]. However, the characteristics of the bacterial communities in the NEW GIFT strain of farmed Nile tilapia, especially the mucosa-associated microbiota, remains scarce. Here, we first characterized the composition and function of the bacterial microbiota in the gill contents, gastrointestinal contents, and mucosae, and we explored the correlation between intestinal microbiota and metabolites in NEW GIFT Nile tilapia under commercial aquaculture conditions. A recent study explored the distribution of the intestinal microbiota in Nile tilapia from two lakes [32], but the wild tilapia was not the new tilapia strain we studied and was not farmed under commercial aquaculture conditions. The functional differences of the bacterial communities at different sites of tilapia were not clearly elucidated, and the correlation between specific intestinal microbes and metabolites was undetermined. Our results provided new insights into the symbiotic microbiota of NEW GIFT Nile tilapia and highlighted the correlation between intestinal microbes and metabolites. Furthermore, understanding the correlation between specific microbes and intestinal metabolites can provide new ideas to improve the health and productivity of this commercially valuable fish species.

To date, the distinct distribution of the microbial communities in different gastrointestinal segments has already been reported in various aquatic animals, including shrimp [33], Atlantic salmon [34], Siberian sturgeon [35], and finless porpoise [36]. Obvious differences exist in the microbial composition and structure of different tissues (skin, gill, and intestine) and digesta (feces) of fish [37]. In this study, the microbiota structures of four gastrointestinal sites were clearly separated, as shown in previous studies on wild tilapia [32] and other animals [33,34,35,36]. Such significant differences in microbial structure led to significant differences in microbial function, which could be attributed to the functional heterogeneity of the gastrointestinal tract, because the stomach is the main site of diet fermentation and the hindgut plays an important role in nutrient absorption [38]. No significant differences were found in the bacterial community structure and composition between the stomach and gill mucosae, which might have been due to the similarity of mucosal niches between these two sites [10]. In addition, three mucosa sites had no significant differences in terms of microbial diversity and displayed a lower microbial diversity compared with the digesta. Our results were consistent with those in previous studies, in that no significant differences in microbial diversity were found at different fish gastrointestinal mucosa sites [34,39]. Moreover, results from other studies showed that several fish had significantly lower richness and diversity for the gastrointestinal mucosa-associated microbiota compared with the digesta-associated microbiota [34,40], which was consistent with our results. The mucosal epithelial cells of fish can secrete immune factors that interact with the mucosal symbiotic microbiota, thereby shaping and restricting the colonization of the microbes [41]. Thus, only a fraction of the bacteria in the intestinal digesta have the characteristics necessary for colonizing the mucosa of fish [41]. This finding might be important to explain the low mucosal microbial diversity. However, a recent study revealed that no significant difference was found in the alpha and beta diversities of the bacterial communities between the intestinal content and intestinal mucosae of wild tilapia in lakes [32]. This divergence was likely due to differences in the lake’s natural water and farmed water environments, as well as fish food sources. Regarding the digesta, the stomach contents displayed a significantly higher microbial diversity compared with the intestinal contents, which was consistent with previous studies on wild tilapia [32], finless porpoises [36], and *Rhabdophis subminiatus* [39]. An explanation may be ascribed to feeding behavior, in which the diet has a high microbial diversity due to its exposure to the water environment containing diverse microorganisms [42], thus resulting in a high microbial diversity in the stomach digesta. Different gastrointestinal regions have distinct physicochemical conditions such as the pH value, redox potential, oxygen concentration, and availability of nutrients [43]. Therefore, many microorganisms in the stomach contents, especially aerobic microbes, cannot survive after entering the intestine, thereby significantly decreasing the microbial diversity of intestinal contents.

The present study was carried out to increase our knowledge of the bacterial microbiota in the gill and gastrointestinal tract of farmed Nile tilapia. The predominant phylum belonged to Proteobacteria in the gill and gastrointestinal tract of tilapia, which was consistent with current consensus of high levels of Proteobacteria in fish [19]. Furthermore, the relative abundance of Proteobacteria in the mucosa samples was significantly higher than that in the digesta samples. The dominant Proteobacteria may play a vital role in the mucosa’s microbial barrier. Moreover, the Proteobacteria phylum contains a variety of opportunistic pathogens, and their existence may contribute to the stimulation of the development of immune system and to the maintenance of normal immune function [2]. As important sites of microbial symbiosis, the gill and stomach have received less attention compared to the intestine. The present results showed that the gill microbiota were dominated by Proteobacteria, followed by Actinobacteria and Firmicutes, and the findings were generally consistent with results from previous studies on other fish species [7,9]. These bacterial phyla were ubiquitous in the gills of fish. Bacteroidetes reportedly participates in carbohydrate transport and protein metabolism, which are significant for digesting diet [44]. Both Cyanobacteria and Chloroflexi can produce energy through photosynthesis [45]. Verrucomicrobia plays a critical role in polysaccharide degradation [46]. In this study, compared with the mucosa samples, the relative abundance values of Bacteroidota, Cyanobacteria, Verrucomicrobia, and Chloroflexi were significantly enriched in the digesta samples, especially in the stomach contents. This divergence might imply that these microbes were derived from the environment (such as feed and water) but cannot colonize the mucosae in great quantities due to the control of the components of the host immune system [3]. Notably, the relative abundance of Firmicutes present in the intestinal content samples was significantly higher than in the stomach content samples. Microbes within the Firmicutes phylum can produce short-chain fatty acids that can provide nutrition for the intestinal mucosal cells [47]. The enrichment of Firmicutes in the intestine contributes to the maintenance of the normal function of the intestinal mucosa and the regulation of the intestinal microecological environment [43,47]. The relative abundance of Actinobacteria in the intestinal contents was much higher than that in other sites. Actinobacteria can produce various potent antibiotics that can inhibit the growth of the intestinal pathogenic bacteria [48]. Therefore, the enrichment of Actinobacteria might be beneficial to intestinal health.

The gill and stomach mucosae shared the core bacterial genera, including *Sphingomonas*, *Ralstonia*, unclassified Comamonadaceae, and *Pelomonas*, all of which were aerobic microorganisms [49,50,51,52]. This finding suggested that both the gill and stomach mucosae had high oxygen levels, which was consistent with adequate exposure to the environment at the two sites [10]. However, further study is required to explain the similarity between the gill and stomach mucosal microbiota observed in the present study. *Sphingomonas* is a Gram-negative bacterium whose cell membrane is composed of sphingolipids [51]. *Sphingomonas* can tightly adhere to cell monolayers and interact with epithelial cells [51]. Human or mammalian T cells can recognize *Sphingomonas* and induce an immune response [53]. *Methylobacterium* may be an important genus that can protect the host against pathogens in fish skin [54]. *R. gilardii* has been recognized as an opportunistic pathogen that can lead to infections, especially in immunocompromised humans [55]. Therefore, the enrichment of *S. aquatilis*, *Methylobacterium*, and *R. gilardii* in the gill and stomach mucosae might play vital roles in stimulating host mucosal immunity and antagonizing pathogen colonization. The relative abundance values of *R. pickettii*, *Pelomonas* (affiliated with Comamonadaceae), and *Staphylococcus* in the mucosa samples were significantly higher than those in the digesta samples. It was generally consistent with recent results on Nile tilapia, which showed that *Ralstonia* and *Pelomonas* were the core microbes in the gut and that their relative abundance values in the mucosae were higher than those in the digesta [32]. *Ralstonia*, *Pelomonas*, and *Staphylococcus* were common bacterial genera of human mucosa-attached microbiome [56,57]. *Ralstonia* was a dominant gut mucosal microbe in cultured sea bass *Dicentrarchus labrax* [58]. It has been reported that *R. picketti* can produce various enzymes, including toluene monooxylase [59], lipase [60], and depolymerase [61]. *Pelomonas* is reportedly involved in various metabolic pathways [62]. Species of *Staphylococcus* can utilize important cellular metabolites of fatty acids [63]. Therefore, the three microbes might play an important metabolic role in the mucosal niches of the host.

Regarding the intestinal content samples, the genera *Undibacterium* and *Escherichia*-*Shigella* were dominant in the intestinal mucosae, and their relative abundance values were significantly higher than those in the intestinal contents. *Undibacterium* is a genus of typically aquatic bacteria and has been found in various sources of freshwater environments [64,65]. Additionally, *Undibacterium* has been described in the intestinal microbiota of shrimp, zebrafish, and bats [66,67,68]. Several species affiliated with the genus *Undibacterium* reportedly produce various fatty acids and polar lipids [64,65]. Thus, the dominant *Undibacterium* in tilapia intestines might be involved in lipid metabolism. This inference was confirmed by our results on intestinal metabolites, i.e., *Undibacterium* was positively correlated with 9,12-(Z,Z)-octadecadienoic acid, glycerol, and methyl-inositol. *Escherichia*-*Shigella* is a common opportunistic pathogen in the gastrointestinal tract of fish [18]. Dietary changes resulted in the decrease of *Escherichia*-*Shigella* in the gut mucosa of the bullfrog, accompanied by a profound reduction of growth performance and immune function [69]. Though some species within *Aeromonas* genus are potential pathogens, *Aeromonas* have been found in the intestinal mucosa of healthy fish [11,40]. Our results showed that the genus *Aeromonas* was significantly more abundant in the intestinal mucosa than in other sites, which was consistent with previous results [40]. *Aeromonas* can produce xylanase and cellulase [11]. *Escherichia*-*Shigella* and *Aeromonas* were significantly enriched in the intestinal mucosae than those in other sites. The genus *Romboutsia* is characterized by the predominance of producing straight-chain saturated and unsaturated fatty acids [70]. Species of the genus *Romboutsia* are reportedly adapted to the environment in the small intestines [71]. Thus, *Romboutsia* in the hindgut might be involved in the biosynthesis of straight-chain fatty acids. These mucosa-attached microbes might play a vital role in the maintenance of the normal physiological functions of host mucosae, including mucosal immunity and metabolism.

Comparing two types of digesta, Chitinophagaceae, Verrucomicrobiae, Comamonadaceae, and chloroplast were significantly enriched in the stomach contents, whereas *Undibacterium*, *Paeniclostridium* and *Candidatus Competibacter* were significantly enriched in the intestinal contents. Most of the dominant taxa in the stomach contents were aerobes or facultative anaerobes, including OTUs belonging to Chitinophagaceae, Verrucomicrobiae, and Comamonadaceae [72,73,74]. *Paeniclostridium* was the third most abundant genus in the intestinal content samples, which was the dominant microbe in the intestinal tract of bats [66]. The genus *Cetobacterium* was abundant in the intestinal content samples, which was consistent with the results of previous studies on tilapia [14,75]. Notably, compared with the gastrointestinal mucosae, *Cetobacterium* was significantly enriched in the gastrointestinal contents. *Cetobacterium* is widely distributed in the intestinal tract of freshwater fish [76]. It can produce vitamin B12 and promote carbohydrate metabolism [76,77]. Since vitamin B12 is a modulator of intestinal microbial ecology [78], the abundance of *Cetobacterium* in the intestinal tract might suggests that studied tilapias had healthy intestinal microbiota. Additionally, species within the genus *Cetobacterium* can promote the decomposition of consumed organic debris, phytoplankton, or zooplankton [79]. Therefore, the enrichment of *Cetobacterium* in the gastrointestinal contents might contribute to tilapia’s digestive function. Together with the abovementioned differences in the structure and diversity, all these microbial divergences revealed the niche differentiation at the organ scale of tilapia’s microbiota [37].

Diet is among the main factors affecting the structure and composition of the intestinal microbiota of genetically improved farmed tilapia [80]. To date, many nutritional factors reportedly affect the structure of host intestinal microbiota, including amino acid [81], fat [82], fructose, and glucose [83]. Though the effects of these nutritional factors on the structure of fish intestinal microbiota have received widespread attention [1], the correlation between specific intestinal microbes and metabolites remains largely unknown. Nutrient is a double-edged sword for the fish’s gut microbiome. Some nutrients can shape healthy gut microbiota, but others may cause an imbalance of gut microbiota [1]. Thus, identifying the interactions between key nutrients and specific microbes may be important for the success of predictive intervention effects. Short-chain fatty acids (SCFAs) are reportedly key bacterial metabolites [47]. SCFAs were not found in the intestinal contents of tilapia, which was consistent with previous results [84]. Lactic acid was the predominant metabolite in the intestinal contents of tilapia, which was consistent with previous results [84]. Lactic acid is an intermediate of glycolysis and can regulate immune response and intestinal mucosal tissue regeneration [85]. Lactic acid was reportedly negatively correlated with the relative abundance values of *Romboutsia*, *Methylocaldum*, *Desulfobacca*, and *Methylocystis*. The lower relative abundance of *Romboutsia* in the intestinal contents compared with the intestinal mucosae might be related to the high concentration of lactic acid in the intestinal contents. The gut is essential for absorbing glucose, and inadequate absorption of glucose ultimately affects fish growth [84]. Glucose showed a significantly negative correlation with the relative abundance values of *Alsobacter*, *Acinetobacter*, *Cyanobium* PCC-6307, and *Turicibacter*, thereby suggesting that these four microbes might have a negative effect on the glucose metabolism in the gut of tilapias. The Spearman correlation heatmap indicated that some metabolites were significantly and positively correlated with the relative abundance values of certain genera but negatively correlated with the relative abundance values of other genera. This phenomenon showed that the metabolites produced by the former might have an inhibitory effect on the latter, suggesting that intestinal metabolites might drive the structure of the bacterial community and regulate the population competition in a direct or indirect way [86].

There are many possibilities in different directions for the correlation between intestinal microorganisms and metabolites. For instance, the relative abundance of *Syntrophus* was negatively correlated with N-acetylglutamic acid, 4-aminobutyric acid, octadecanoic acid, and arachidonic acid but positively correlated with cysteine, aspartic acid, and methionine. Some metabolites were not significantly correlated with microorganisms because enzymes that produced these metabolites were encoded in organisms [18]. Interestingly, the relative abundance values of some potentially beneficial bacteria, including *Undibacterium*, *Crenothrix*, and *Cetobacterium* were positively correlated with most intestinal metabolites. The genera *Undibacterium* and *Cetobacterium* may be involved in the intestinal metabolism, as described above. *Crenothrix* may be involved in methane oxidation and contribute to methane removal in the gut [87]. These results suggested that these potentially beneficial microbes might play important roles in promoting nutrient digestion and metabolism in the gut. We also observed that the relative abundance values of *Mycobacterium*, *Desulfomonile*, *Escherichia*-*Shigella*, *Paeniclostridium*, *Aeromonas*, *Clostridium sensu stricto* 1, *Alsobacter*, *Acinetobacter*, *Cyanobium* PCC-6307, and *Turicibacter* were negatively correlated with most intestinal metabolites. Most of these genera were potential opportunistic pathogens, such as *Acinetobacter*, *Mycobacterium*, *Escherichia*-*Shigella*, *Paeniclostridium*, *Aeromonas*, and *Clostridium sensu stricto* 1 [66]. Therefore, we hypothesized that an increased proportion of potentially beneficial microbes might contribute to an improvement of intestinal metabolic function, whereas an increased proportion of opportunistic pathogens might impair intestinal metabolic function. The dietary supplementation of probiotics or prebiotics can reshape the gut microbiota of fish [88]. Our experiment revealed the close correlation between specific intestinal microbes and metabolites, which might contribute to screening targeted probiotics or prebiotics and adding them to the feed to regulate the digestion and metabolism of nutrients in the intestinal tract. However, this study did not reveal a causal relationship between metabolites and gut microbiota.

The combination of numerous intestinal microbes in the intestinal contents with the corresponding metabolites provided favorable data support for the evaluation of the function of intestinal microbiota [86]. Our results showed that intestinal metabolites were related to the metabolism of carbohydrates, amino acids, lipids, cofactors, and vitamins, which was generally consistent with the dominant metabolic pathways in the predicted function of the intestinal microbiota. In addition, approximately 66% of the predictive function in the gastrointestinal microbiota was related to metabolic pathways, especially carbohydrate and amino acid metabolisms, which supported the previous consensus that the fish gut microbiota may play important roles in host nutrient metabolism [1]. Such results showed the presence of consistency between the composition or function of intestinal microbiota and intestinal metabolites. Notably, the predictive function of digesta-associated microbiota was significantly different from that of the mucosa-associated microbiota. Compared with the mucosa-associated microbiota, four metabolic pathways—especially global and overview maps—were significantly enriched in the digesta-associated microbiota. This finding suggested that digesta-associated microbiota play a more important role in nutrient metabolism than mucosa-associated microbiota. On the contrary, the pathway for xenobiotics biodegradation and metabolism in the mucosa-associated microbiota was significantly more abundant than those in the digesta-associated microbiota. This might suggest that the mucosa-associated microbiota have a strong capacity for degrading exogenous pollutants or toxic substances, thus protecting mucosal tissues from exogenous damage. Microbial functional differences at different sites of tilapia were revealed, thereby further elucidating the unique function of symbiotic microbiota at different sites. However, the PICRUSt2 remains limited in identifying consistent, differentially abundant functions [30]. Therefore, further metagenomic studies should be conducted to identify the unique function of symbiotic microbiota at different sites.

## 5. Conclusions

This study is the first to comprehensively characterize the taxonomic and functional profiles of the gill and gastrointestinal microbiota in the NEW GIFT strain of adult farmed Nile tilapia. It was found that the microbial diversity, structure, and predictive function were significantly different across gastrointestinal regions and sample types. *Pelomonas*, *R. pickettii*, Comamonadaceae, and *Staphylococcus* were significantly enriched in the mucosa samples, whereas many bacterial taxa were significantly enriched in the digesta samples, including Chitinophagaceae, *Cetobacterium*, *Candidatus Competibacter*, HOC36, *Methyloparacoccus*, and chloroplast. Additionally, *Undibacterium*, *Escherichia*-*Shigella*, *Paeniclostridium,* and *Cetobacterium* were prevalent in the intestinal content and mucosa, whereas *S. aquatilis* and *R. gilardii* were commonly found in the gill and stomach mucosae. Interestingly, the relative abundance values of several potentially beneficial microbes were positively correlated with most intestinal metabolites, whereas the relative abundance values of some potential opportunistic pathogens were negatively correlated with most intestinal metabolites. However, the potential function of these related microorganisms and their regulation mechanisms regarding host intestinal metabolism need to be further studied.

## Figures and Tables

**Figure 1 microorganisms-09-00617-f001:**
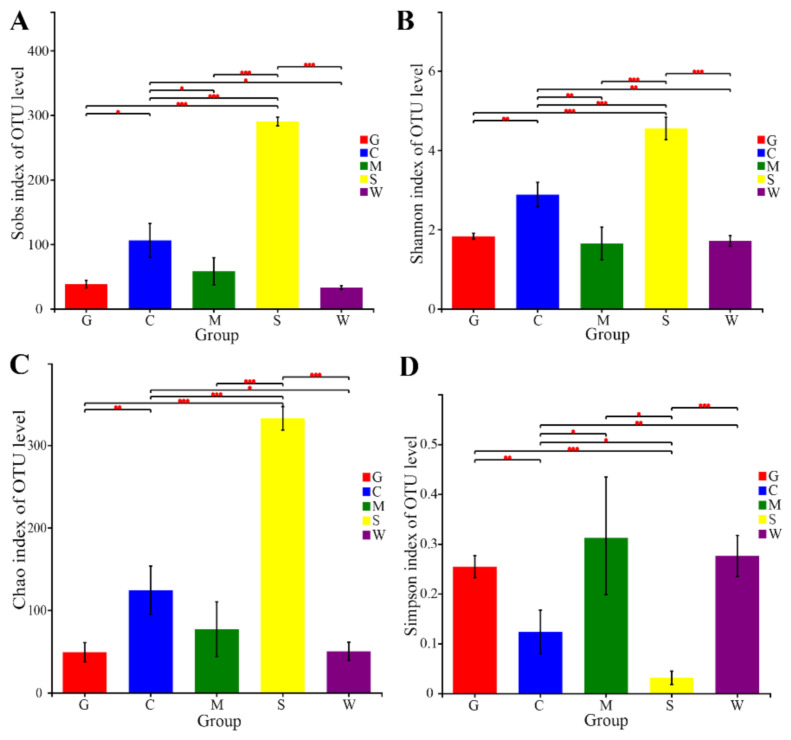
Comparison of alpha diversity indices of the bacterial communities at different sites. (**A**) Comparison of the Sobs index of the bacterial communities at different sites, (**B**) comparison of the Shannon index of the bacterial communities at different sites, (**C**) comparison of the Chao index of the bacterial communities at different sites, and (**D**) comparison of the Simpson index of the bacterial communities at different sites. Higher Sobs and Chao values indicate a higher richness; higher Shannon and lower Simpson values indicate a higher diversity. Statistical significances between two sites were considered at * *p* < 0.05, ** *p* < 0.01, and *** *p* < 0.001 by Welch *t*-test. G: gill mucosae (G1–G4); C: intestinal contents (C1–C4); M: intestinal mucosae (M1–M4); S: stomach contents (S1–S4); W: stomach mucosae (W1–W4).

**Figure 2 microorganisms-09-00617-f002:**
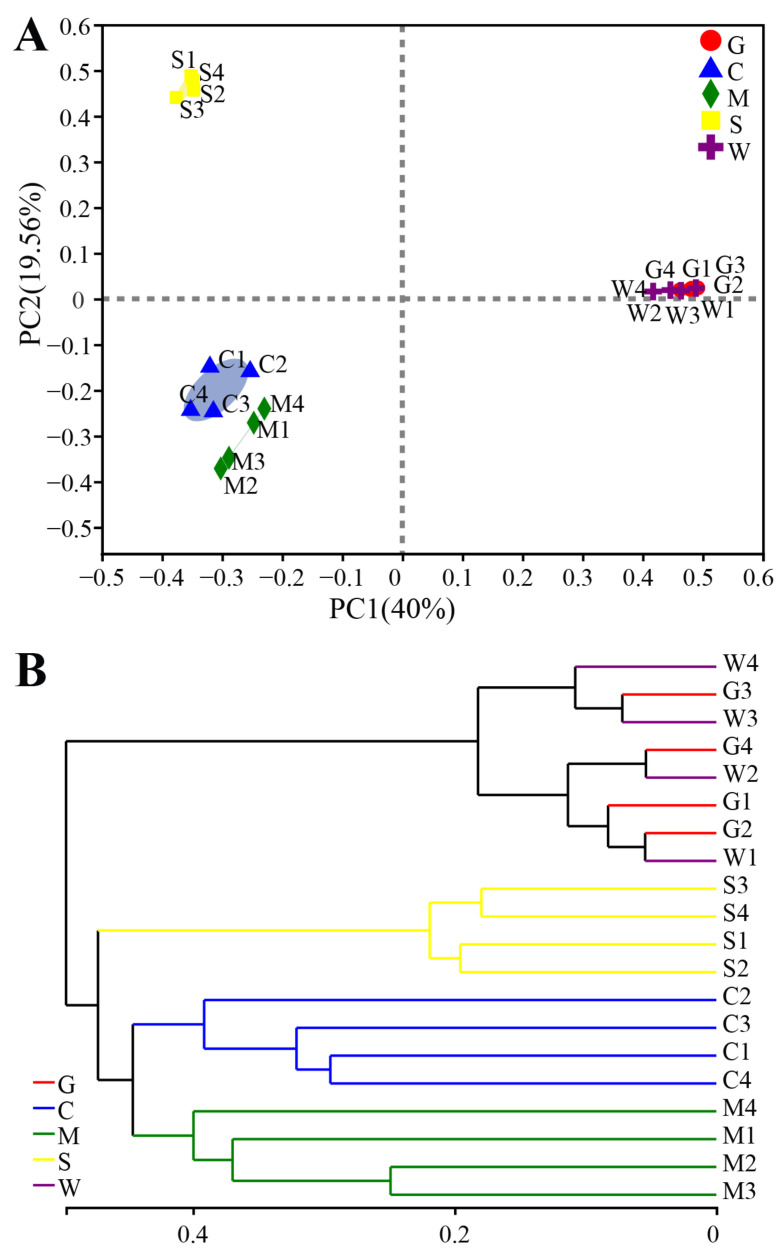
Analysis of the bacterial community structure at different sites. (**A**) Principal coordinate analysis based on the Bray-Curtis metric of the bacterial communities. The percentages indicate the relative contribution of the principal components. (**B**) The hierarchical clustering tree based on Bray-Curtis metric of the bacterial communities. G: gill mucosae (G1–G4); C: intestinal contents (C1–C4); M: intestinal mucosae (M1–M4); S: stomach contents (S1–S4); W: stomach mucosae (W1–W4).

**Figure 3 microorganisms-09-00617-f003:**
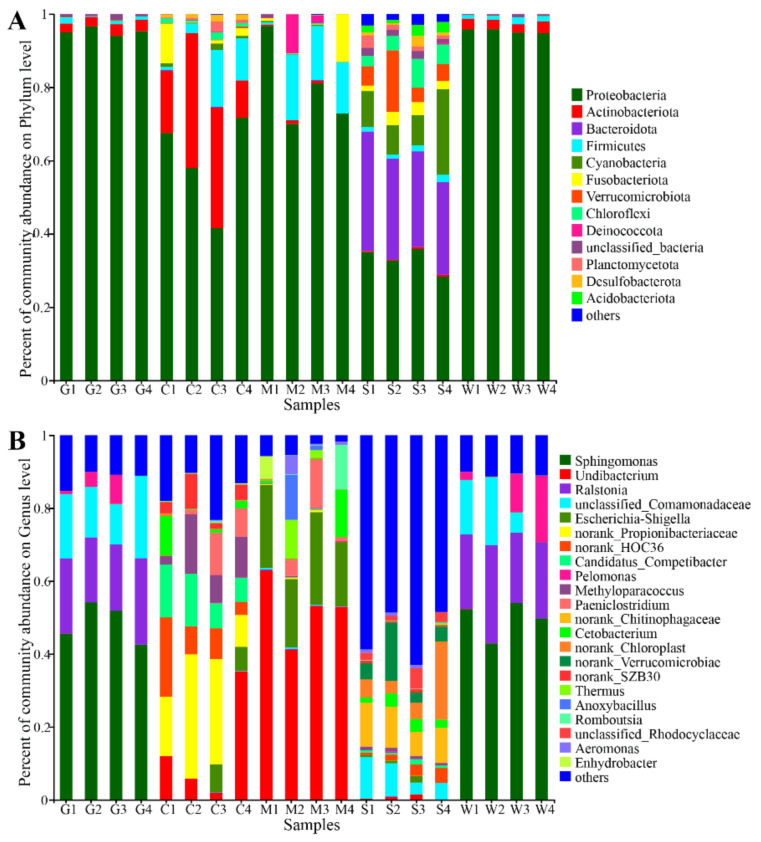
Distribution of the bacterial communities in all twenty samples at (**A**) the phylum level or (**B**) the genus level. G: gill mucosae (G1–G4); C: intestinal contents (C1–C4); M: intestinal mucosae (M1–M4); S: stomach contents (S1–S4); W: stomach mucosae (W1–W4).

**Figure 4 microorganisms-09-00617-f004:**
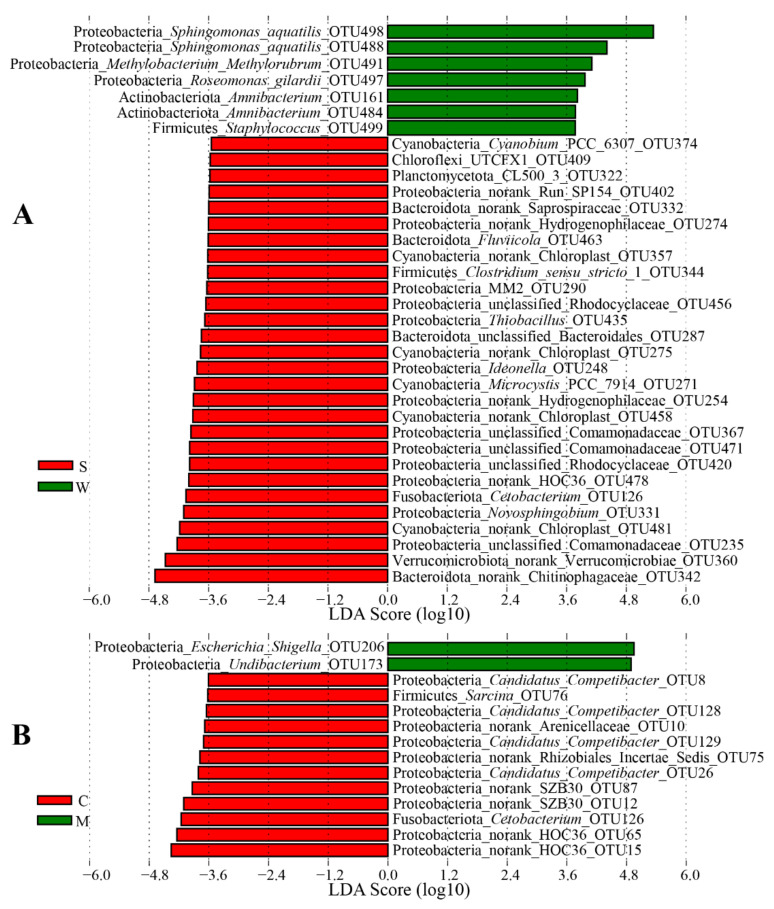
(**A**) Linear discriminant analysis (LDA) effect size (LEfSe) showing differences in the bacterial communities at the operational taxonomic unit (out) level between the stomach contents and stomach mucosae. (**B**) LEfSe showing differences in the bacterial communities at the OTU level between the intestinal contents and intestinal mucosae. The highlighted taxa are enriched in the group that corresponds to each color. LDA scores can be interpreted as the degree of difference in the relative abundance of OTUs. S: stomach contents (S1–S4); W: stomach mucosae (W1–W4); C: intestinal contents (C1–C4); M: intestinal mucosae (M1–M4).

**Figure 5 microorganisms-09-00617-f005:**
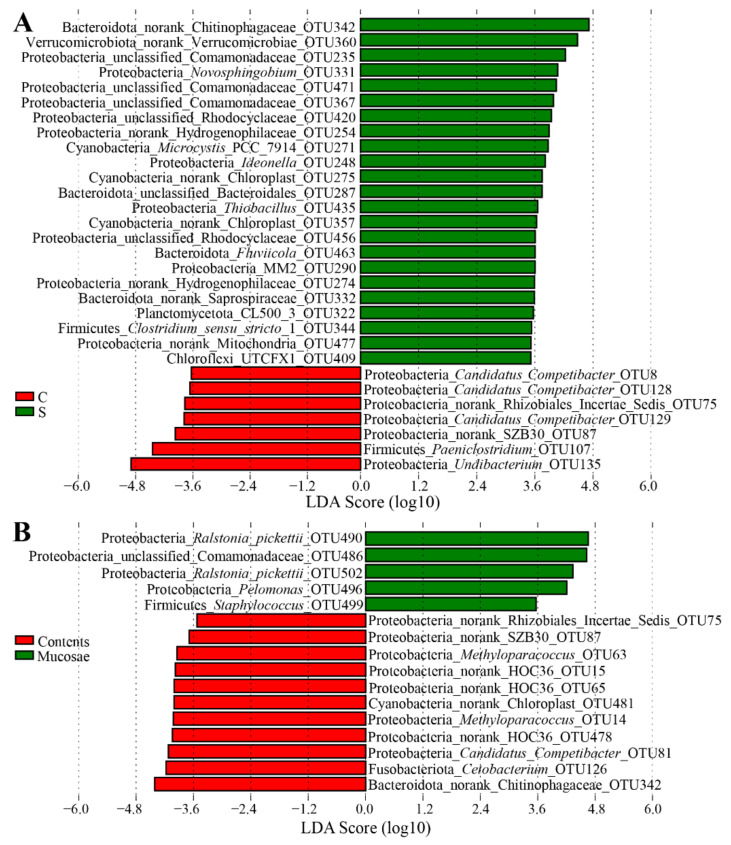
(**A**) LEfSe showing differences in the bacterial communities at the OTU level between the stomach contents and contents. (**B**) LEfSe showing differences in the bacterial communities at the OTU level between the content (C and S) and mucosa (G, M, and W) samples. The highlighted taxa are enriched in the group that corresponds to each color. LDA scores can be interpreted as the degree of difference in the relative abundance of OTUs.

**Figure 6 microorganisms-09-00617-f006:**
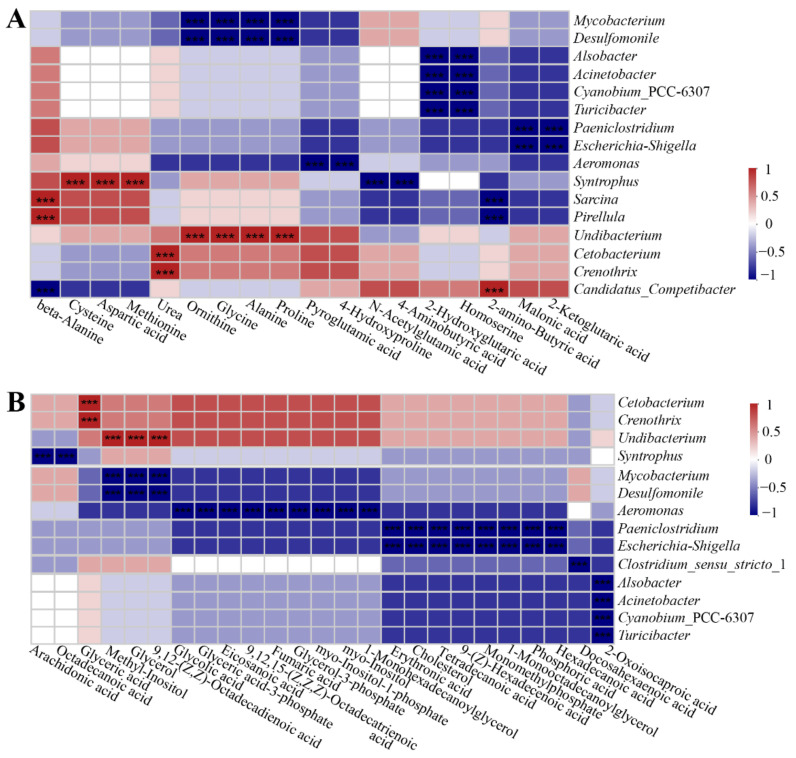
The significant correlation between intestinal main bacterial genera and metabolites. (**A**) The correlation of intestinal bacteria and amino acid-related metabolites. (**B**) The correlation of intestinal bacteria and lipid-related metabolites. The correlation coefficient is represented by different colors (red: positive correlation; blue: negative correlation). * Represents significantly negative or positive correlations (*** *p* < 0.001).

**Figure 7 microorganisms-09-00617-f007:**
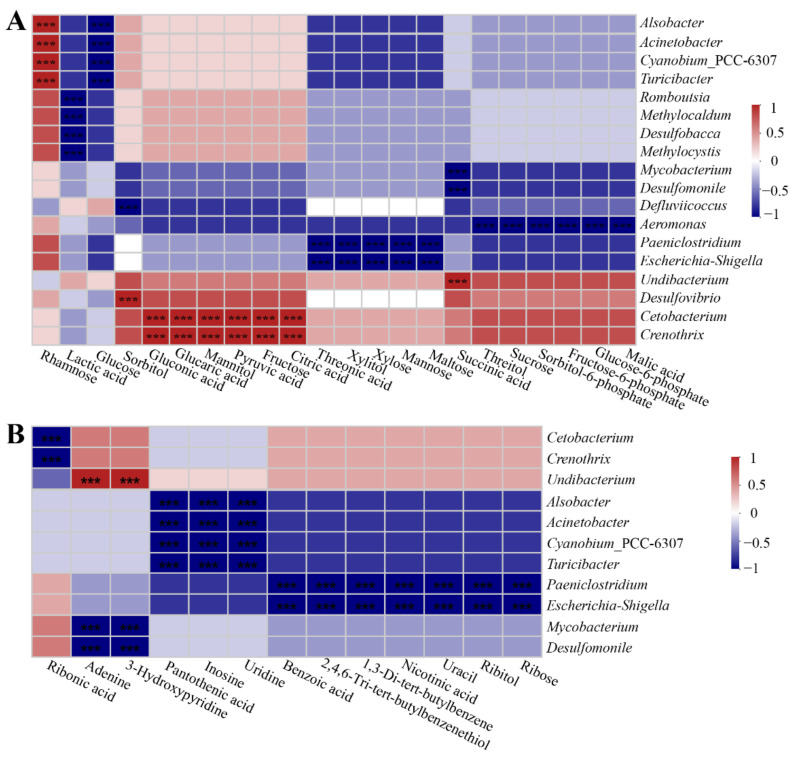
The significant correlation between intestinal main bacterial genera and metabolites. (**A**) The correlation of intestinal bacteria and carbohydrate-related metabolites. (**B**) The correlation of intestinal bacteria and other metabolites. The correlation coefficient is represented by different colors (red: positive correlation; blue: negative correlation). * Represents significantly negative or positive correlations (*** *p* < 0.001).

**Figure 8 microorganisms-09-00617-f008:**
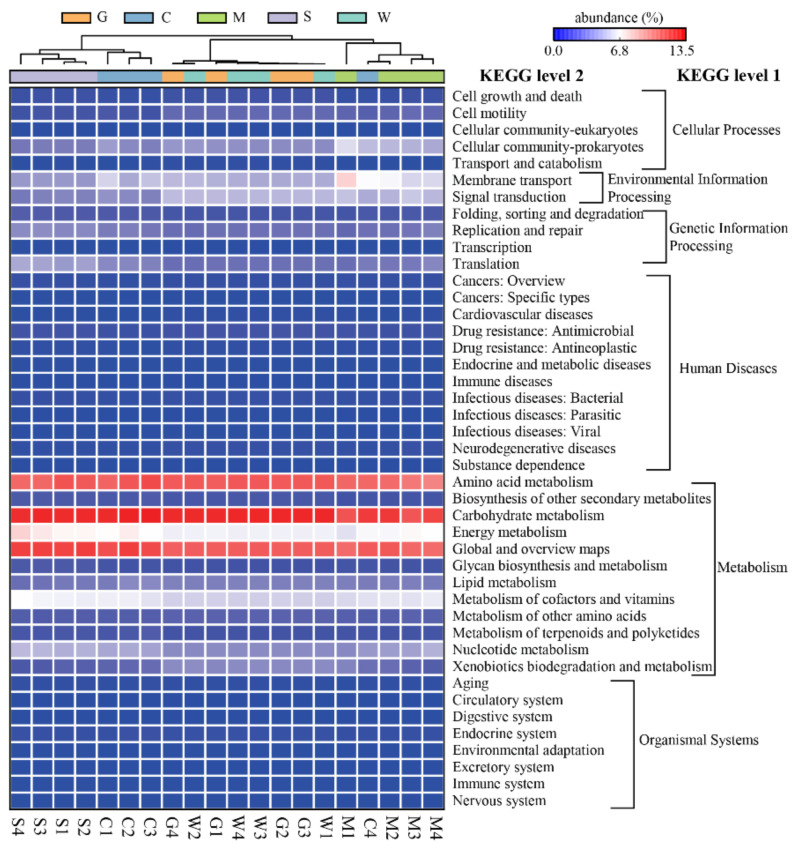
Heatmap profiles showing the functional categories (Kyoto Encyclopedia of Genes and Genomes (KEGG) level 1 and level 2) of the bacterial communities, as predicted by the Phylogenetic Investigation of Communities by Reconstruction of Unobserved States (PICRUSt2) analysis. The functional clustering analysis is based on the unweighted pair-group method with arithmetic means (UPGMA). Rows represent the KEGG Orthology (KO) functions, columns represent the 20 samples, and the color intensity in the heatmap represents the relative abundance (%) of the functional categories. G: gill mucosae (G1–G4); C: intestinal contents (C1–C4); M: intestinal mucosae (M1–M4); S: stomach contents (S1–S4); W: stomach mucosae (W1–W4).

**Figure 9 microorganisms-09-00617-f009:**
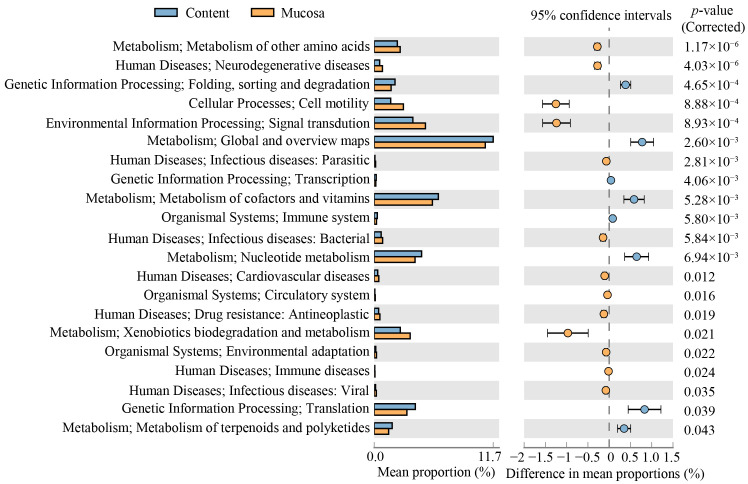
Extended error bar showing the differentiated putative functions of the bacterial communities between the content and mucosa samples as predicted by PICRUSt2 analysis. Rows represent the 21 differentiated KEGG Orthology (KO) functions (corrected *p* < 0.05), and the bars in the graph represent the mean proportion (%) of the functional categories. The 95% confidence intervals reflect the difference in mean proportions (%), and corrected *p*-values are displayed on the right of the figure.

## Data Availability

The raw sequencing data were deposited in the NCBI Sequence Read Archive under accession number PRJNA692658 (http://www.ncbi.nlm.nih.gov/sra, accessed on January 17, 2021).

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
