# Peer review of "Taxonomic and Functional Characteristics of the Gill and Gastrointestinal Microbiota and Its Correlation with Intestinal Metabolites in NEW GIFT Strain of Farmed Adult Nile Tilapia (Oreochromis niloticus)"

_microorganisms, 2021, doi:10.3390/microorganisms9030617_

Round 1

Reviewer 1 Report

Dear authors, I have read your work and is interesting. There are some points that I think you need to change in order to improve your article.

Introduction

you say a lot the words "of fish". Please change or delete these words. You do not need to repeat them all the time.

References - You have 2 choices - Either you change some references with others so you do not repeat too many times the same reference (e.g. ref. 8 - 5 times in the same paragraph) or the second choice - rewrite the sentences in order to cite just one/two times the same reference. Make this process to all your introduction.

Line 173-182 - Make this a separate paragraph. It is the most important part of the introduction.

Materials and methods

Add a reference for the 28.5°C. Or state clearly if it was your idea. This is necessary for others to replicate your experiment and to understand why this temperature level.

Results section - you have a balanced section, with a lot of results and a good dimension of their interpretation.

Discussion section - try to not make reference to the previous described results. It is hard for a reader to go back to certain table or figure.

Pay attention to references cited in introduction and do not repeat them many times.

Make the sentences shorter - do not extend them over 3-4 lines.

Conclusion section

Lines 966-969 - these sentences are not necessary. Stay focused on your main results.

Lines 975-983 - Delete "Comparing the two sample types". Make this sentence shorter.

Line 987 - Do not speculate. present your results. You can speculate in the discussion section, but I suggest not.

Overall, in this section you need to have short sentences. I suggest to rewrite it in totally. The reader will lose in a larger text of just one sentence.

Reviewer 2 Report

The Microrganisms_1114197 manuscript takes into consideration the gill and intestine microbiota in a farmed tilapia strain.

The paper represents a new piece of knowledge of the microbial populations present on the mucous membranes of fish. The work represents a new piece of knowledge of the microbial populations present on the mucous membranes of fish.

The introduction is a little too complex and can definitely be compressed. For the rest, the materials and methods and results are fully described and incorporated into the text. The discussion is very broad and well detailed, taking into account the characterization carried out and all the possible implications and the various correlations.

For this reason, in my opinion, the paper is well structured and clear enough to be published after minor revision.
